# Films of Chitosan and *Aloe vera* for Maintaining the Viability and Antifungal Activity of *Lactobacillus paracasei* TEP6

**Carolina Barragán-Menéndez, Didiana Gálvez-López****, Raymundo Rosas-Quijano, Miguel Salvador-Figueroa, Isidro Ovando-Medina and Alfredo Vázquez-Ovando ***

Instituto de Biociencias, Universidad Autónoma de Chiapas, Boulevard Príncipe Akishino sin número Colonia Solidaridad 2000, Tapachula CP 30798, Mexico; carvy_2705@hotmail.com (C.B.-M.); didiana.galvez@unach.mx (D.G.-L.); raymundo.rosas@unach.mx (R.R.-Q.); miguel.salvador@unach.mx (M.S.-F.); isidro.ovando@unach.mx (I.O.-M.)
* Correspondence: jose.vazquez@unach.mx; Tel.: +52-9626427972

**Abstract:** The present study aimed to evaluate the effect of *Aloe vera* addition on the viability and antifungal activity of TEP6 (*Lactobacillus paracasei*) bacteria immobilized on chitosan films for 28 days. Different chitosan and *A. vera* proportions and carbon sources at several pH values were tested as formulations for supporting the microorganism. Bacterial viability was maintained in freshly made films, with values of 10.4, 10.8 and 10.9 log CFU·g$^{-1}$ for the formulations containing 70% (T11), 100% (T8) and 100% (T16) of *A. vera*, respectively. The same formulations (T8, T11 and T16) maintained bacterial viability for 14 days of film storage with a loss to values of 9.5 log CFU·g$^{-1}$. By applying a quarter fraction $2^{5-2}$ experimental design with an array of five factors, the factors with the greatest influence on viability and antifungal activity were determined. The optimal conditions for viability were the formulation with 100% *A. vera*, pH 4.5 and 0.1 M glucose. The antifungal activity of fresh films was influenced by the formulation with 10 g·L$^{-1}$ glycerol and 100% *A. vera*, showing a 60% inhibition of fungal (*Colletotrichum gloeosporioides*) growth. The films developed in this study may have the potential to be used as coatings on vegetal products susceptible to attack by *Colletotrichum gloesporioides*.

**Keywords:** *Lactobacillus paracasei*; *Colletotrichum gloeosporioides*; MTT; glycerol; inhibition

## 1. Introduction

Lactic acid bacteria (LAB) are a group of bacteria widely used in the food industry. One of their most common uses is in the bioconservation of foods because they inhibit the growth of different microorganisms, including bacteria, yeast and fungi, through the production of substances such as hydrogen peroxide, bacteriocins and organic acids (e.g., lactic acid and acetic acid) [1]. The majority of related studies have focused on the use of LAB as controllers of bacteria that cause food spoilage [2] and/or that cause food poisoning [3].

In the last two decades, several studies have demonstrated the ability of LAB to inhibit the growth of fungi that cause food spoilage, as in the case of *Penicillium expansum* [4] and other phytopathogenic fungi [5,6]. Recently, the isolation of the LAB *Lactobacillus paracasei* TEP6 from tepache was reported, and it was demonstrated to possess strong antifungal activity against the fungus *Colletotrichum gloeosporioides*, which causes anthracnose in papaya [7].

Depending on the nature of the food, LAB are incorporated in different ways to perform their protective activity. The method of incorporation is crucial to maintain the viability and functionality of

these microorganisms; the cells are immobilized in polymer matrices [8]. A recently explored method is immobilization in edible films [9,10] because these films are applied to various foods, including fresh fruits. When developing cell entrapment matrices, the nature of the components should be considered in such a way that they can act as a source of nutrients for their maintenance of LAB [11,12]. Such films can be made with distinct biomolecules (proteins, polysaccharides and lipids) alone or in combination [13].

In this regard, chitosan is one of the most widely used polysaccharides for film production [14]. Chitosan, a partially deacetylated derivative of chitin, is a hetero-polysaccharide composed of 2-amino-deoxy-β-D-glucopyranose and 2-acetamido-deoxy-β-D-glucopyranose (chitin) residue and its films have good properties, i.e., transparency, resistance, flexibility, efficiency to gas exchange and harmlessness to humans and other animals [14,15] Chitosan-based edible coatings can also be used as carriers of food ingredients (antimicrobials, texture enhancers, nutraceuticals, cells ) to improve the safety, quality and functionality of the fruit and vegetables [15].

*Aloe vera* gel is composed of polysaccharides, glycoproteins, vitamins, enzymes and phenolic compounds [16]. Due to its composition, it has been reported to provide the necessary nutrients to LAB and has even been shown to promote the growth of *L. acidophilus*, *L. plantarum* and *L. casei* when added to the culture medium [17]. Another study reported that extracts of *Aloe* gel increased the cell growth of *L. acidophilus* [18]. However, to date, there is no information regarding the use of *A. vera* gel as a substrate for LAB immobilized in chitosan films or the effect of this combination on the antimicrobial capacity of LAB.

Therefore, the objective of the present study was to evaluate the viability and antifungal activity against the phytopathogenic fungus *C. gloeosporioides* of LAB TEP6 immobilized in films formulated with a combination of chitosan, *A. vera* gel and other carbon sources.

## 2. Materials and Methods

### 2.1. Reagents

Chitosan low molecular weight from shrimps' shells with 75%–85% deacetylation and MW = 50,000–190,000 Da was obtained from Sigma-Aldrich®. Other reagents included 85% lactic acid (Meyer), glycerol (Meyer), lactose (Bioxon), anhydrous glucose (Meyer), 3-(4,5-dimethylthiazol-2-yl)-2,5-diphenyltetrazolium bromide (MTT) (Sigma), dimethyl sulfoxide (DMSO) (Sigma), dextrose agar (ADP) (Sigma-Aldrich) and MRS agar (BD Difco™). All the chemicals were of reactive grade.

### 2.2. Bacterial Strain

Strain TEP6 was previously isolated from tepache samples and was identified as *L. paracasei* and was selected for its ability to inhibit the phytopathogenic fungus *C. gloeosporioides* [7]. The strains were reseeded in MRS agar and a cell suspension was prepared with MRS broth. Bacterial counts in the suspension were done by the most probable number method (MPN) and correlated with the optical density (OD) at 560 nm, after performing a calibration curve. When required, cells were cultivated under agitation until an OD of 2.25 was achieved, which corresponded to $2.8 \times 10^9$ CFU mL$^{-1}$.

### 2.3. Aloe vera Gel

Mature *A. vera* leaves (1.33 Brix) without visual damage were collected. They were washed in water and disinfected by immersion in a 1% (*v/v*) solution of sodium hypochlorite for 15 min. *A. vera* gel was obtained according to a previously reported method [19]. The parenchyma was separated from the cortex and homogenized in a blender at low speed for 5 min. The obtained liquid was passed through a sieve to remove the fiber. This process was performed at the time the gel was used to avoid oxidation.

## 2.4. Cell Viability in Chitosan–Aloe vera Films

To investigate the effect of the *A. vera* concentration and pH on the viability of LAB, films were formulated with different proportions of chitosan: *A. vera* gel solution. The initial concentration of the chitosan solution in each formulation was varied to maintain a fixed final concentration (1.5% *w/v*) in all cases. These ratios (formulations) were evaluated at two different pH values. The 16 resulting formulations are shown in Table 1.

**Table 1.** Formulations with different ratios of chitosan (Ch): *Aloe vera* gel (A).

| Formulation | Ratio of Ch: A | pH | Formulation | Ratio of Ch: A | pH |
|---|---|---|---|---|---|
| T1 | 90:10 | 4.5 | T9 | 90:10 | 5.5 |
| T2 | 80:20 | 4.5 | T10 | 80:20 | 5.5 |
| T3 | 70:30 | 4.5 | T11 | 70:30 | 5.5 |
| T4 | 60:40 | 4.5 | T12 | 60:40 | 5.5 |
| T5 | 50:50 | 4.5 | T13 | 50:50 | 5.5 |
| T6 | 25:75 | 4.5 | T14 | 25:75 | 5.5 |
| T7 | 100:0 | 4.5 | T15 | 100:0 | 5.5 |
| T8 | 0:100 | 4.5 | T16 | 0:100 | 5.5 |

The film solutions were elaborated following the procedures described by Monzón-Ortega et al. [19]. First, chitosan was dissolved in sterile water containing 1.5% lactic acid (*v/v*) by magnetic stirring. The corresponding amount of *A. vera* gel was then added, and the pH was adjusted with 0.1 N NaOH according to the formulation.

To incorporate the bacteria, the required volume of MRS broth with cells to achieve a concentration of 10.4 log CFU·g$^{-1}$ of film was centrifuged at 5880× *g* for 20 min, and the cell pellet was washed twice in 1 mL of PBS. The cells were then incorporated into the chitosan: *A. vera* solutions. After homogenization, 3 mL of solution was poured into sterile Petri dishes (60 mm in diameter) and allowed to stand until the formation of the film (48 h on average).

The viability of the cells incorporated into the films at the time of film formation (storage time 0) was evaluated following the methodology based on the tetrazolium reduction test, known as the MTT assay [20]. Viability (%) was calculated as follows: Viability = (OD viable cells from films/OD of initial number cells) × 100. The results were transformed to be presented as log CFU·g$^{-1}$ of film.

The films from the formulations that showed higher viability values were reprocessed in the same manner as described above except that the viability was evaluated for 14 days every 24 h by the method described above.

## 2.5. Effect of Film Composition on Cell Viability and Antifungal Capacity

To verify the contribution of the film components to the viability and antifungal capacity, a quarter fraction $2^{5-2}$ experimental design was followed. The eight formulations and evaluated factors are shown in Table 2. Five repetitions of each formulation were made. The films were prepared with the same concentration of chitosan and lactic acid previously described, and the following factors were investigated: (1) concentration of *A. vera* gel, (2) concentration of glycerol, (3) presence or absence of glucose, (4) presence or absence of lactose and (5) pH of the solution. The inclusion of glucose or lactose as carbon source was made in previous literature reporting that the supplementation of media with carbohydrates may increase the bioactivity of cells [21]. As response variables, the cell viability was measured by the previously described procedure every 7 days (for 28 days). In parallel, every 24 h for 7 days, the antifungal capacity of the films was measured following the methodology previously described [22]. The fungus *C. gloeosporioides* was used for this purpose. This strain came from infected papaya fruits and has been identified by dichotomous keys [23]. Fungal spores were sown and cultivated in malt extract agar (MEA) pH 7. Mycelial disks of the fungus (6 mm in diameter) were cut and placed in the center of Petri dishes containing MEA, and disks of films 10 mm in diameter

were placed on top of the fungus. The plates were incubated at room temperature, and mycelial growth was measured every 24 h. From these data, the percentage of growth inhibition (I) was calculated by the equation I = [(Dc − Dt)/Dc] × 100, where Dt is the growth diameter of the formulation mycelium and Dc is the growth diameter of the control mycelium (mycelium in MEA without any film). For each formulation, five repetitions were performed.

**Table 2.** A quarter fraction $2^{5-2}$ experimental design was used to investigate the effect of factors influencing the viability and antifungal capacity of films containing the TEP6 strain.

| Formulation | Glycerol | Lactose | Glucose | pH | *Aloe vera* |
|:---:|:---:|:---:|:---:|:---:|:---:|
| 1 | 1 | −1 | −1 | −1 | −1 |
| 2 | −1 | −1 | −1 | 1 | 1 |
| 3 | −1 | 1 | −1 | −1 | 1 |
| 4 | −1 | −1 | 1 | 1 | −1 |
| 5 | −1 | 1 | 1 | −1 | −1 |
| 6 | 1 | −1 | 1 | −1 | 1 |
| 7 | 1 | 1 | 1 | 1 | 1 |
| 8 | 1 | 1 | −1 | 1 | −1 |

The numbers (−1, 1) represent the following conditions: 10 g·L⁻¹ glycerol = −1, 20 g·L⁻¹ glycerol = 1; absence of lactose = −1, 0.1 M lactose = 1; absence of glucose = −1, 0.1 M glucose = 1; pH 4.5 = −1, pH 5.5 = 1, 30% *A. vera* gel = −1, 100% *A. vera* gel = 1.

## 2.6. Data Analysis

The software Statgraphics Centurion XV v. 15.2.06 was used to evaluate the experimental design. Analysis of variance and subsequent comparisons were performed by the Tukey test ($p < 0.05$), and contribution analysis (standard effect) of the evaluated factors was carried out.

## 3. Results and Discussion

### 3.1. Cell Viability in Fresh Chitosan–Aloe vera Films

Figure 1 shows the viability of TEP6 incorporated into the films formulated with different concentrations of *A. vera* and chitosan immediately after the preparation (storage day 0). T8 formulation (100% *A. vera*, pH 4.5) and T16 (100% *A. vera*, pH 5.5) had the greatest viability, with values of 10.8 and 10.9 log CFU·g⁻¹, respectively, surpassing the initial cell value (10.4 log CFU·g⁻¹). These formulations were significantly equal ($p > 0.05$) to each other but different ($p < 0.05$) from the rest of the formulations. Both formulations had the same composition (100% *A. vera*) and only differed in their pH, suggesting that, when using *A. vera* as carrying agent for the microorganism, pH does not influence its viability. These results coincide with those of other studies [18] that showed that when using 5% gel extracts of *A. barbadensis* and *A. arborescens*, the growth of the bacterium *L. acidophilus* was induced. However, in the same study, LAB *L. delbrueckii* did not show the same behavior, and its growth was reduced. This result suggests that the *L. paracasei* (TEP6) bacterium used in the present study is not sensitive to the metabolites contained in the *A. vera* gel; on the contrary, it efficiently uses the polysaccharides of the gel (mostly acemannan and glucomannan), as reported by Nagpal and Kaur [24]. These authors attributed the maintenance of high viabilities and even the growth of LAB (*L. plantarum*, *L. casei* and *L. helveticus*) to the use of some substrates present in gels (inulin, oligofructose, lactulose and raftilose) as prebiotics.

The viability results of this study contrast with those reported by others [17], who assert that the growth of LAB (*L. acidophilus*, *L. plantarum* and *L. casei*) can be affected by *A. vera* juice concentrations greater than 25% when incorporated into the culture medium. The origin of LAB TEP6 or the genetic variant may be a possible explanation for the observed differences, because the bacteria were isolated from a fermented product and could have been better adapted to the metabolites present in the *A. vera* gel. Another formulation that maintained initial viability was T11 (70% chitosan and 30% *A. vera*, pH

5.5); however, this formulation was significantly equal ($p > 0.05$) to most formulations. Formulations T2, T4 and T12 displayed lower viability, with values less than 10 log CFU·g$^{-1}$ (Figure 1). Both formulations where 40% *A. vera* was incorporated (T4 and T12) have the lowest cell viability values; which is difficult to explain from the perspective of the composition of the films.

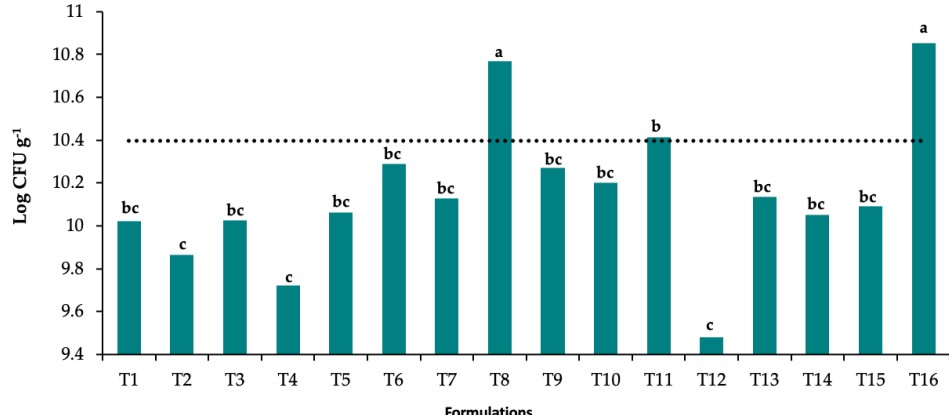

**Figure 1.** Viability of the TEP6 bacteria incorporated into films freshly prepared with different concentrations of *Aloe vera* and chitosan. The dotted line represents the initial number of cells added in solution (10.4 log CFU·g$^{-1}$); the bars represent the number of cells in the different formulations of freshly processed dry films. The letters represent the analysis of variance.

### 3.2. Cell Viability in Films Stored for 14 Days

The viability of the LAB in films stored for 14 days at room temperature for formulations T8, T11 and T16 is shown in Figure 2. From day 8, no significant differences were observed ($p > 0.05$) between formulations T8 and T16; however, formulation T11 showed significantly lower ($p < 0.05$) viable cells from the other formulations for almost all days of storage. Except for TR11 formulation (from day 0 to 1), the atypical variations between days into formulations is due to normal variability for the experimental procedure since no significant differences among days were revealed by the statistical analysis (data not shown). The trend observed during the 14 days of storage was similar to that observed in the freshly prepared films (Figure 1).

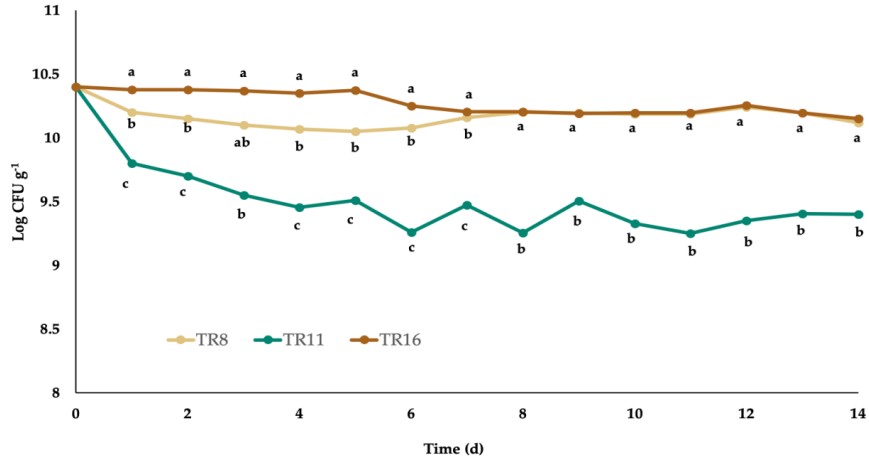

**Figure 2.** Viability of the TEP6 bacteria incorporated into films stored at room temperature. TR8: (100% *A. vera*, pH 4.5), TR16: (100% *A. vera*, pH 5.5), TR11: (70% chitosan and 30% *A. vera*, pH 5.5). The letters represent the analysis of variance to observe the differences between each formulation per day.

TR8 and TR16 maintained viability over the 14 days, while for TR11 the viability decreased to 9.8 log CFU·g$^{-1}$ on the first day and was maintained throughout the storage until recording a value

of 9.4 log CFU·g$^{-1}$ on the final day of storage. The similarity in the results for formulations TR8 and TR16 again reveals that pH does not seem to be a factor that affects viability because both formulations contained 100% *A. vera* and differed only in their pH values. The similarity of the two pH values (4.5, 5.5) could be an explanation for this, which agrees with other authors [11], who stated that acidic pH values (below five) can cause slight stress in cells, thereby increasing viability as a response mechanism. Likewise, these authors also proved that the use of prebiotics can result in an increase in the survival rate of LAB in acidic conditions, which coincides with the results of the present study and demonstrates that the *A. vera* gel would function as a prebiotic for TEP6. Ai et al. [25] also agree that LAB viability is higher at acidic pH values (4.2) than at basic pH values. It has been reported that basic pH favors the accumulation of lactate, which can result in the loss of both membrane integrity and metabolic activity. The interaction between the pH and the chemical composition of the culture medium also seems to be important for cell viability, as others reported that survival is higher when LAB are cultivated at basic pH values (7.4 and 8.5) in MRS medium [26]; however, when the same cells were cultured in brain and heart infusion (BHI) broth, pH 5.5 was optimal to maintain cell viability.

### 3.3. Effect of Film Composition on Cell Viability and Antifungal Capacity

When analyzing the individual effects of the factors (components of films) on viability, it was found that the addition of lactose and glycerol had no significant effect on viability (Figure 3) for most of the days analyzed; the addition of 20 g·L$^{-1}$ glycerol was only significant for viability on day 14 of storage. On the other hand, the use of 100% *A. vera* was found to significantly increases viability, which coincides with previous experiments where the formulations that showed the greatest viabilities were from those with 100% *A. vera*. However, in contrast to what was found in the previous stages, a pH of 4.5 did have an effect on the viability on days 7, 14 and 28.

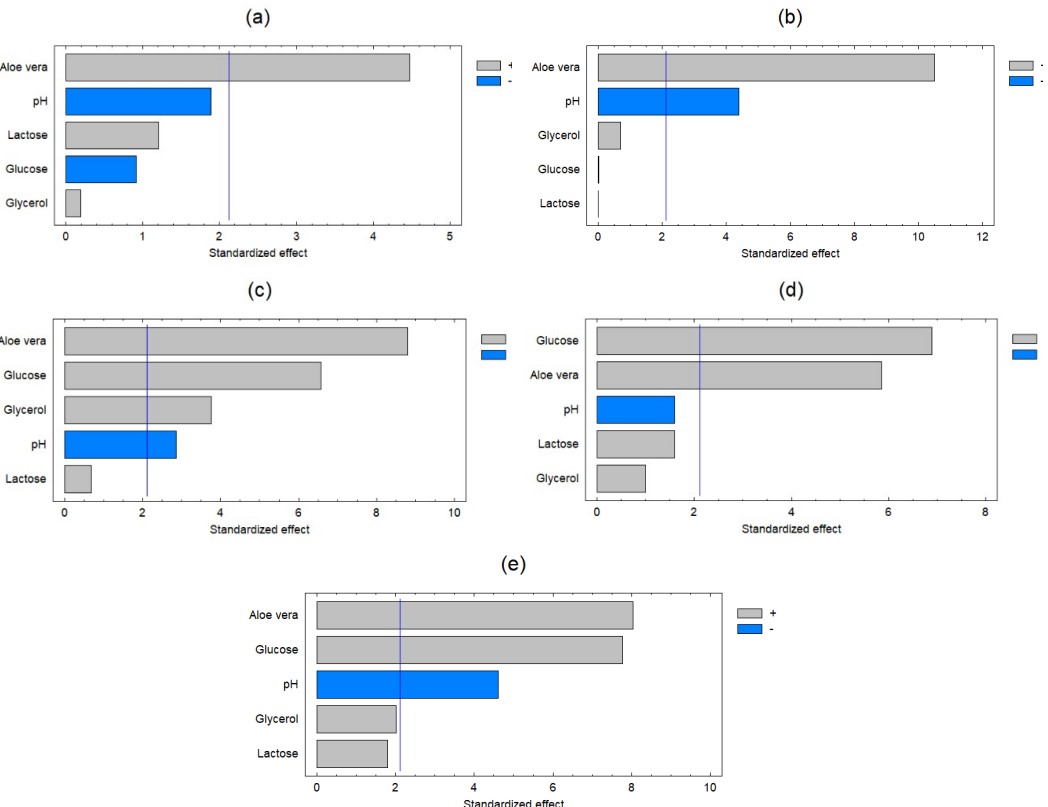

**Figure 3.** Pareto charts of the standardized effect of different factors on the viability of the TEP6 bacteria incorporated into films and stored for 28 days, evaluated on days 0 (**a**), 7 (**b**), 14 (**c**), 21 (**d**) and 28 (**e**).

Additionally, the addition of 0.1 M glucose had a positive effect on viability on days 14, 21 and 28 of film storage. This result coincides with that reported by others [27], who reported that 10 g·L$^{-1}$ glucose and 10 mL·L$^{-1}$ formic acid resulted in increased viability of *L. rhamnosus* R-2002 at 48 h. Various carbon sources have been shown to stimulate both the growth and antifungal activity of LAB. Commonly used carbon sources are glucose, lactose, sucrose, fructose and maltose. However, glucose is the most often used because many studies report that this monosaccharide is easier to assimilate and allows an increased production of enzymes and metabolic compounds that promote an increase in biomass and, consequently, the production of antimicrobial substances, compared to sucrose and fructose [28].

### 3.4. Effect of Factors on the Antifungal Capacity of the TEP6 Bacteria Immobilized in Freshly Prepared Films

Analyzing the individual effects of the factors (components of the films) on the antifungal capacity of the TEP6 bacteria in freshly dried films (0 days of storage) revealed that the addition of 10 g·L$^{-1}$ glycerol had the strongest effect on the antifungal capacity during the seven days of fungal growth (Figure 4). This finding coincides with other authors [27], who also suggested that glycerol has an adjuvant effect on antifungal activity because it can reduce the surface hydrophobicity of cells, thereby increasing the antifungal effect of other metabolites. This effect occurred only partially in the present study. Although glycerol was a significant factor ($p < 0.05$) for several days (Figure 4), the antifungal activity in the best case was 59% in T3 (containing 10 g·L$^{-1}$ glycerol) (Figure 5), while in the other formulations containing glycerol (T2-T5), no such response was observed.

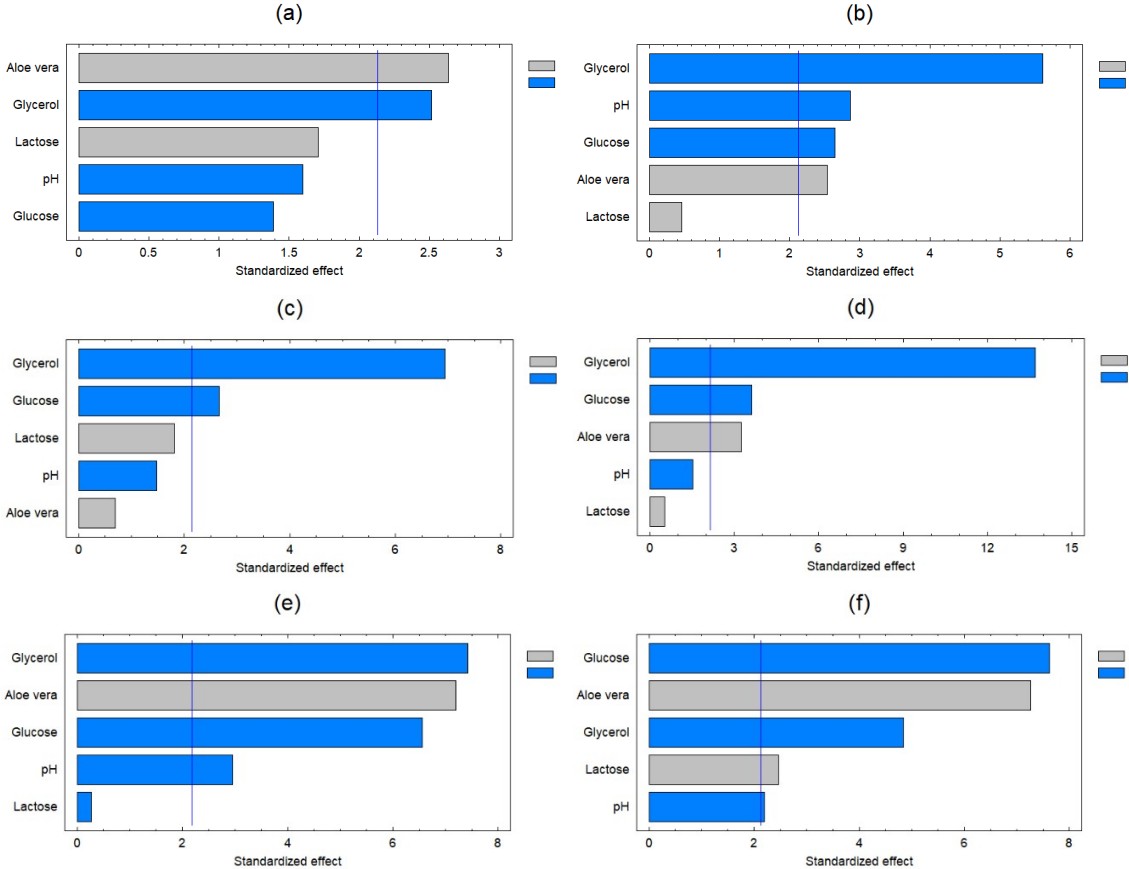

**Figure 4.** *Cont.*

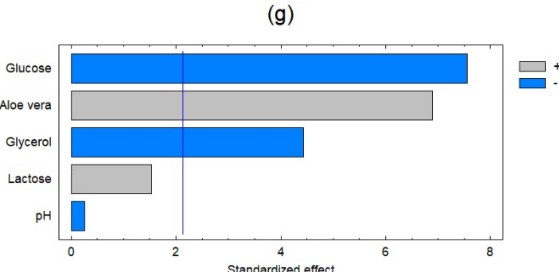

**Figure 4.** Pareto charts of the standardized effect of different factors on the antifungal capacity of the TEP6 bacteria incorporated into films with 0 days of storage, evaluated on days 1 (**a**), 2 (**b**), 3 (**c**), 4 (**d**), 5 (**e**), 6 (**f**) and 7 (**g**).

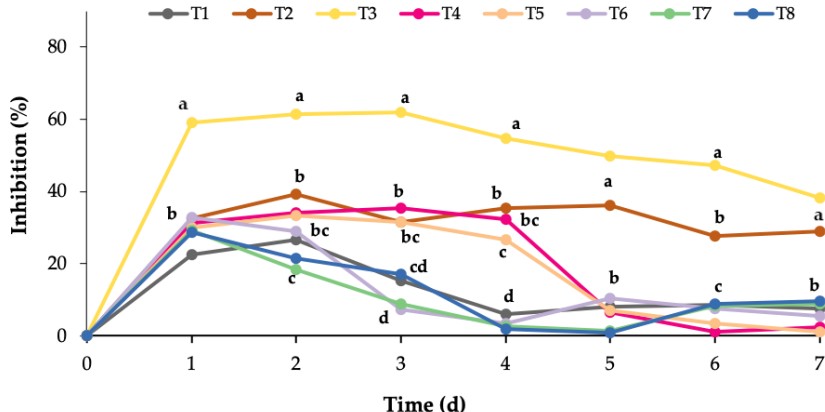

**Figure 5.** Inhibition (%) of the fungus *Colletotrichum gloeosporioides* by the TEP6 bacteria incorporated into freshly processed films (storage day 0) and evaluated for 7 days. The letters represent the analysis of variance to observe the differences between each formulation per day.

The use of 100% *A. vera* gel also had an effect on days 1.5 and 7, which coincides with the above viability results. On the other hand, not adding glucose also had an effect on almost every day of evaluation, except on day one. The pH and lactose were not significant factors to maintain or increase the antifungal capacity of the films. In contrast, it has been reported that the gene expression of antimicrobial metabolites (mainly bacteriocins) is regulated by pH [26]. The pH values evaluated in the present study (4.5 and 5.5) are possibly not high enough to show an effect of pH on genetic regulation; the cited authors evaluated a wide pH range (4.5, 5.5, 6.2, 7.4 and 8.5) and reported that the optimal production of bacteriocins was at pH 6.2.

However, in general, for the antifungal capacity, optimal inhibition values were not obtained even from the start of film storage (Figure 5), because the formulations showed an average value of 40% inhibition of phytopathogen fungal growth, and only T3 reached 59% inhibition with fresh films (storage day 0). In addition, freshly made films presented a better visual aspect (Figure 6) than stored ones.

Toward the end of fungal development (day seven), only formulations TR2 and TR3 maintained inhibition values above 29%, and these formulations were significantly different from each other ($p > 0.05$) and different ($p < 0.05$) from the other formulations. These results agree with the findings of Bravo-De la Cruz et al. [29], who found that the antifungal capacity of *L. plantarum* in chitosan films was maintained at 100% inhibition only on the day of film preparation (day 0 of storage), but the inhibition capability drastically declined thereafter. However, unlike the results from these authors, the TEP6 bacteria did not reach 100% growth inhibition of *C. gloeosporioides* in any of the formulations. These results were lower than expected according to the potential of LAB TEP6. Another study has

reported that this bacterium exerts strong inhibition (100%) on both spore germination and mycelium growth in *C. gloeosporioides* [7].

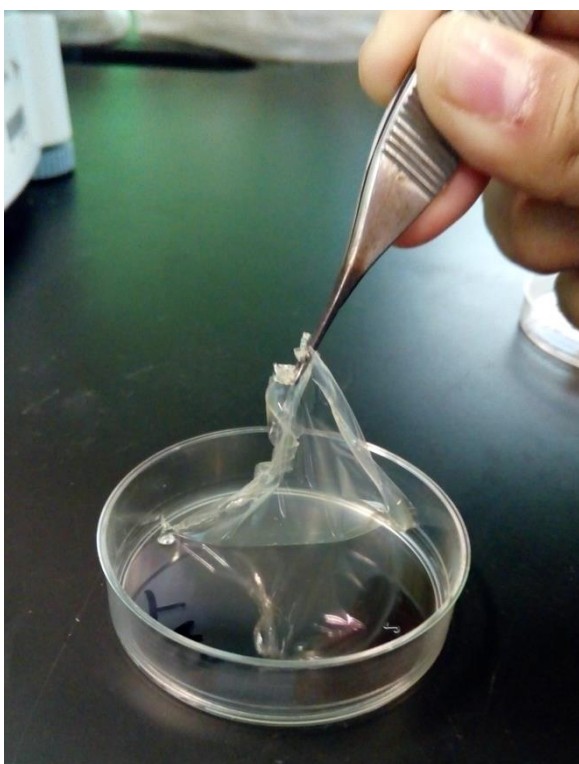

**Figure 6.** Freshly made films from T3 formulation containing *L. paracasei* TEP6 cells.

Therefore, the antifungal activity of films stored for more than seven days was not evaluated. Bravo-De la Cruz et al. [29] reported that once LAB trapped in chitosan films lose their antifungal capacity, they do not recover it again during storage at room temperature. Our results could be confirming that hypothesis, and show that chitosan exerts an antagonistic effect against LABs; although there are reports of Gram-positive bacteria containing teichoic acid and lipoteichoic acid that are poly-anionic surface polymers, interacting with intracellular substances, so that the vital bacterial activities of chitosan are impaired [15].

## 4. Conclusions

The use of *A. vera* gel maintained the viability of *L. paracasei* TEP6 cells trapped in chitosan films for up to 28 days. When *A. vera* was used exclusively, an increase in the number of cells in the newly formed films was also observed. The highest viability values were obtained when the films were formulated with 100% *A. vera* gel at pH 4.5 and with a concentration of 0.1 M glucose. The highest inhibition of the fungus *C. gloeosporioides* reached values close to 60% in the formulation where $10 \text{ g·L}^{-1}$ glycerol was added to the film made of 100% *A. vera* gel. Although this antifungal activity is acceptable, other factors or conditions that could increase the antifungal capacity of the elaborated films should be evaluated.

**Author Contributions:** A.V.-O. conceived and design the study; C.B.-M., I.O.-M. and R.R.-Q. performed the experiments; A.V.-O., M.S.-F. and I.O.-M. provided the materials and resources; A.V.-O. and M.S.-F. supervised the experiment; C.B.-M., D.G.-L. and A.V.-O. analyzed the data; C.B.-M. and A.V-O. wrote the paper; all authors revised the manuscript. All authors have read and agreed to the published version of the manuscript.

**Funding:** This research was partially funded by SEP-Mexico through the program PROFOCIE 2018-2019 and The APC was partially funded by PRODEP-SEP-Mexico through the program "support for publication expenses".

**Conflicts of Interest:** The authors declare no conflict of interest.

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
