# Peer review of "Films of Chitosan and Aloe vera for Maintaining the Viability and Antifungal Activity of Lactobacillus paracasei TEP6"

_coatings, doi:10.3390/coatings10030259_

Round 1
Reviewer 1 Report
The manuscript “Biofilms of Chitosan Enriched with Aloe vera for Maintain the Viability and Antifungal activity of Lactobacillus paracasei TEP6” deals with the study of the viability and antifungal ability of a lactic acid bacterium incorporated into chitosan films containing Aloe vera.
Regarding the title, formulations with only A. vera are presented as those showing the best results therefore this formulations are not “chitosan films enriched with A. vera” but “A. vera films”. Also pay attention to the use of gerund after the preposition for.
Based on this I suggest the following title “Biofilms (or films) of Chitosan and Aloe vera for Maintaining the Viability and Antifungal activity of Lactobacillus paracasei TEP6”.
I would not use the term “biofilm” neither in the title nor in the rest of the manuscript because is preferentially employed when referring to the aggregation of microorganisms in a naturally produced biopolymeric matrix. Nevertheless, I will leave such decision in authors’ hands.
From my practical experience working with the incorporation of living microorganisms into edible coatings/films formulations, chitosan would not be the most appropriate carrier because of its antimicrobial properties.
Some suggestions regarding English style are given in the attached file in order to avoid excessive repetition of expressions and to improve the quality and clarity of the text but, in general, English grammar is correct.
Likewise, some suggestions are given for the Results and Discussion that could help to enrich this section (see attached document).

Reviewer 2 Report
Dear Authors,
this is very interesting and well written article. please see my comments below:
46-51. more references and facts about chitosan films are needed.
Introduction part is short and more details about chitosan and technology are desired.
- type, grade of chitosan and raw material origin?
the results are well presented however the pictures of biofilms and materials produced would be very beneficial for this article!
BR
Round 2
Reviewer 1 Report
I am thankful to the authors for considering the given suggestions. The English grammar has been considerably improved although some minor mistakes have been detected so some suggestions have been given below. The authors have also improved the quality of the figures and some key aspects of the discussion that I think have importantly increased the overall quality of the manuscript. Based on these observations, I consider the manuscript is adequate for its publication in Coatings after the minor corrections suggested.
Line 2: re-write as “… for Maintaining the Viability…”
Abstract: Move the sentence “Different chitosan and A. vera proportions and carbon sources at several pH values were tested as formulations for supporting the microorganism” right after the first sentence (after 28 days).
Introduction:
Line 52: use the plural “… its films have…”
Line 55: “of fruit and vegetables”.
Line 76: use the plural “Bacterial counts were done…”
Line 117-118: It is great that you added the information about the incorporation of glucose and lactose but the sentence is a bit confusing. Suggest re-phrasing “The inclusion of glucose or lactose as carbon sources was made on previous literature reporting that the supplementation of media with these carbohydrates may increase the bioactivity of cells”
Line 153: correct this “to some the use of some substrates”
Line 168-169: suggest re-writing as “… the initial amount of cells”
Line 177: It is good that you re-did the statistical analysis, just to check. Taking this into account I suggest re-phrasing the sentence as “… experimental procedure since no significant differences among days were revealed by the statistical analysis (data not shown)”.
Author Response
The authors gratefully acknowledge the meticulous review made by the reviewer.
We have made all the changes that so timely and accurately issued us.
We think that this version of the manuscript has been improved thanks to its revision effort.